# The GABARAP Co-Secretome Identified by APEX2-GABARAP Proximity Labelling of Extracellular Vesicles

**DOI:** 10.3390/cells9061468

**Published:** 2020-06-16

**Authors:** Julia L. Sanwald, Gereon Poschmann, Kai Stühler, Christian Behrends, Silke Hoffmann, Dieter Willbold

**Affiliations:** 1Institut für Physikalische Biologie, Heinrich-Heine-Universität Düsseldorf, Universitätsstraße 1, 40225 Düsseldorf, Germany; julia.sanwald@uni-duesseldorf.de; 2Institute of Biological Information Processing (IBI-7: Structural Biochemistry), Forschungszentrum Jülich, Leo-Brandt-Straße, 52428 Jülich, Germany; 3Institute of Molecular Medicine I, Heinrich-Heine-Universität Düsseldorf, Universitätsstraße 1, 40225 Düsseldorf, Germany; gereon.poschmann@uni-duesseldorf.de (G.P.); kai.stuehler@uni-duesseldorf.de (K.S.); 4Molecular Proteomics Laboratory, Biologisch-Medizinisches Forschungszentrum (BMFZ), Heinrich-Heine-Universität Düsseldorf, Universitätsstraße 1, 40225 Düsseldorf, Germany; 5Munich Cluster for Systems Neurology (SyNergy), Ludwig-Maximilians-Universität München, Feodor-Lynen-Straße 17, 81377 München, Germany; christian.behrends@mail03.med.uni-muenchen.de

**Keywords:** ATG8, GABARAP, autophagy, APEX2, proximity, extracellular vesicle, exosome, secretion

## Abstract

The autophagy-related ATG8 protein GABARAP has not only been shown to be involved in the cellular self-degradation process called autophagy but also fulfils functions in intracellular trafficking processes such as receptor transport to the plasma membrane. Notably, available mass spectrometry data suggest that GABARAP is also secreted into extracellular vesicles (EVs). Here, we confirm this finding by the immunoblotting of EVs isolated from cell culture supernatants and human blood serum using specific anti-GABARAP antibodies. To investigate the mechanism by which GABARAP is secreted, we applied proximity labelling, a method for studying the direct environment of a protein of interest in a confined cellular compartment. By expressing an engineered peroxidase (APEX2)-tagged variant of GABARAP—which, like endogenous GABARAP, was present in EVs prepared from HEK293 cells—we demonstrate the applicability of APEX2-based proximity labelling to EVs. The biotinylated protein pool which contains the APEX2-GABARAP co-secretome contained not only known GABARAP interaction partners but also proteins that were found in APEX2-GABARAP’s proximity inside of autophagosomes in an independent study. All in all, we not only introduce a versatile tool for co-secretome analysis in general but also uncover the first details about autophagy-based pathways as possible biogenesis mechanisms of GABARAP-containing EVs.

## 1. Introduction

In the past few years, the analysis of the direct protein environment of a protein of interest (POI) in a well-defined cellular compartment, so-called proxitome analysis, has grown into an emerging area of research. This has become possible by the development of proximity-based labelling strategies such as biotinylation by an engineered ascorbate peroxidase (APEX). Initially developed for high-resolution electron microscopy [1], APEX was soon applied to living cells for the first time, determining the proteins inside the mitochondrial matrix and demonstrating that APEX can also be used for proteomic mapping. In order to do so, APEX is tagged to a POI and expressed in a eukaryotic host cell system. When incubating the cells with biotin-phenol and H_2_O_2_, APEX catalyses the oxidation of biotin-phenol to a short-lived biotin-phenoxyl radical, which is unable to cross biological membranes and labels proteins at electron-rich amino acids in close proximity to POIs (<20 nm) [2].

Shortly afterwards, an improved APEX2 variant exhibiting enhanced catalytic efficiency was generated [3], which since then has been applied to map a variety of protein–protein interactions. Amongst others, APEX2 was tagged to a G-protein coupled receptor (GPCR) to track GPCR signalling [4], to the fibroblast growth factor 1 to identify interaction partners on the cell surface [5], and to Rab proteins to determine new Rab functions [6]. Furthermore, APEX2 was used to define the protein environment in other specific cellular compartments like lipid droplets, organelles regulating lipid storage [7], and autophagosomes. For the latter, the APEX2 system was applied to human orthologues of yeast autophagy-related 8 (Atg8), among them γ-aminobutyric acid type A receptor-associated protein (GABARAP) [8]. However, to our knowledge, the APEX2-mediated proximity labelling technique has yet been most commonly applied to adherent cells, leaving open a big area of interest ranging from eukaryotic cells in suspension over unicellular organisms to extracellular vesicles (EVs). One step in this direction was recently taken by Singer-Kruger et al. by developing an APEX2-labelling strategy for *Saccharomyces cerevisiae* cells [9]. For the next step, the main goal of the here-presented study was therefore to test the applicability of APEX2-mediated proximity labelling to EVs (Appendix A).

In this study, we examine EVs in general, as there is often only little information on the EV subtype that is involved in the secretion of the respective POI. The “EV” term includes all kinds of secreted membrane vesicles in the extracellular space, which are highly heterogenous, depending on their cells of origin and their pathways of biogenesis. Commonly, EVs are subdivided into two main groups: microvesicles, which develop by shedding of the plasma membrane, and exosomes, which are formed in an initial step by the invagination of early endosomes. Thereby, multivesicular bodies containing intraluminal vesicles are formed. When fusing with the plasma membrane, the intraluminal vesicles are secreted as exosomes (reviewed by van Niel et al. [10]). Further mechanisms of unconventional protein secretion also involve other vesicle types, for example, secretory lysosomes or even autophagosomes (reviewed by Nickel [11]). Although the content of EVs varies, there are still features that they have in common. For instance, EVs generally contain mRNA, which can be transported to a recipient cell, where the mRNA can be translated and thus serves as intracellular communication [12,13]. Furthermore, it is possible to characterise EVs based on their protein content. Typical EV marker proteins include tetraspanins, e.g., CD81; cytosolic proteins, like members and accessory proteins of the endosomal sorting complexes required for transport (ESCRT) machinery, e.g., ALG-2-interacting protein X (Alix; official gene name: programmed cell death 6-interacting protein (PDCD6IP)); heat shock proteins, e.g., Hsc70; or annexins, e.g., annexin V [14]. Amongst numerous others, these EV marker proteins are usually accessed to determine the quality of an EV sample before conducting a more detailed analysis such as mass spectrometry.

During autophagy, a highly conserved cellular homeostasis mechanism [15], the autophagy-related 8 (ATG8) protein family member GABARAP was shown to be involved in the autophagosome–lysosome fusion process [16] and to be lipidated by a ubiquitin-like system [17,18]. The lipidation does not only support GABARAP’s binding to autophagosomal membranes [19,20], enabling the attachment of both autophagic cargo and their receptors as well as regulators of the core autophagic machinery [21,22]. In fact, by connecting to tubulovesicular structures [23], it likely also promotes the initially described function of GABARAP: the trafficking of receptors to the plasma membrane, for example, the GABA_A_ receptor [24], the human transferrin receptor [25], or the angiotensin II type 1 receptor [26], making it a versatile binding hub. Furthermore, it was shown that GABARAP mediates insulin secretion together with the motor protein kinesin-1 heavy chain (KIF5B) by localising insulin-loaded vesicles at microtubules and enhancing vesicle movement [27]. Despite taking part in all these events, the secretion of GABARAP itself has not yet been studied in detail. However, through a query in Vesiclepedia [28,29], we realised that GABARAP is already listed as an extracellular vesicle (EV) protein in samples from human [30,31,32,33] and mouse [34] cancer cells, and in a recent proteomic study, ATG8-protein family members were detected in EVs from different cell lines [35]. In all the underlying studies linked to the respective entries, GABARAP was found either at the mRNA level or at the protein level by mass spectrometric methods. With our work, we provide further evidence of GABARAP’s secretion, as we reveal its presence in EVs of cell culture supernatants from different human cell lines and human blood plasma by immunoblotting. Finally, we investigated the co-secretome of GABARAP in EVs by applying, for the first time, an APEX2-based proximity labelling strategy in EVs. In this context, our goal was to establish a method that solely includes those EVs in the analysis that contain the POI, in our case GABARAP. The APEX2-based approach, by which only EV proteins are labelled that are localised in the same EV as the POI fused to APEX2, might help to cope with this task and to define a co-secretome. Using this technique, we observed an overlap of the EV GABARAP proxitome with that of the autophagosomal GABARAP proxitome [8], suggesting an autophagy-based pathway as one possible biogenesis mechanism of GABARAP-containing EVs.

## 2. Materials and Methods

### 2.1. Antibodies

The following primary antibodies were used for immunoblotting at a concentration of 1:1000: Alix (Cat. No. sc-49268, Santa Cruz Biotechnology, Dallas, TX, USA), Annexin V (Cat. No. ab14196, abcam, Cambridge, UK), Calnexin (Cat. No. ab22595, abcam), CD81 (Cat. No. sc-23962, Santa Cruz Biotechnology), c-myc (Cat. No. A190-104A, Bethyl Laboratories, Montgomery, TX, USA), GABARAP (Cat. No. 18723-1-AP, Proteintech, Rosemont, IL, USA) and Hsc70 (Cat. No. ab51052, abcam). Streptavidin-HRP (Cat. No. S911, Invitrogen, Carlsbad, CA, USA) was used at a 0.3 µg/mL concentration. Anti-goat (Cat. No. sc-2354, Santa Cruz Biotechnology), anti-mouse (Cat. No. P026002-2, Agilent Technologies, Santa Clara, CA, USA) and anti-rabbit (Cat. No. 31460, Invitrogen) were used as secondary antibodies at a concentration of 1:10,000. The following primary antibodies were used for immunofluorescence: c-myc (Cat. No. A190-104A, Bethyl Laboratories) at a concentration of 1:200 and GABARAP 8H5 (in-house [36], undiluted). The Goat-488 (Cat. No. ab150133, abcam) and rat-Cy5 (Cat. No. 712-175-150, Jackson ImmunoResearch, Cambridgeshire, UK) secondary antibodies were used at a concentration of 1:250.

### 2.2. Cloning

The pcDNA5/FRT/TO myc-APEX2-GABARAP vector was obtained from Christian Behrends’ lab. The vector pcDNA5/FRT/TO myc-APEX2 was generated by introducing two stop codons in the linker region between APEX2 and GABARAP using the QuikChange II XL Site-Directed Mutagenesis Kit (Cat. No. 200521, Agilent Technologies).

### 2.3. Cell Culture and Transfection

Human embryonic kidney 293 (HEK293; Leibniz Institute DSMZ—German Collection of Microorganisms and Cell Cultures, Braunschweig, Germany; DSMZ no. ACC 305) cells and hepatocellular carcinoma (Huh7; COSMIC sample ID 907071) cells were cultured at 37 °C in a humidified 5% CO_2_ incubator in Dulbecco’s Modified Eagle’s Medium (DMEM, Cat. No. D5796, Sigma-Aldrich, St. Louis, MO, USA) supplemented with 10% fetal calf serum (FCS, Cat. No. F9665, Sigma-Aldrich). For human neuroblastoma (SH-SY5Y; Leibniz Institute DSMZ; DSMZ no. ACC 209) cells, DMEM supplemented with 20% FCS was used. All the cells were routinely checked for mycoplasma contamination.

For transient transfection with myc-APEX2 and myc-APEX2-GABARAP constructs, 4–6 × 10^5^ cells were seeded on the previous day into an appropriate vessel: the wells of 6-well culture plates (Cat. No. 10062-892, VWR, Randor, PA, USA) were used for immunoblotting (IB) and MS, while for immunofluorescence (IF), poly D-Lysine coated bottom dishes (Cat. No. P35GC-0-10-C, MatTek Corporation, MA, USA) were used. Transfection was carried out on the next day with 1.0 µg total DNA using PolyFect Transfection Reagent (Cat. No. 301107, Qiagen, Hilden, Germany) according to the manufacturer’s instructions.

### 2.4. Blood Sampling

Human (autologous) blood plasma was donated from J.L.S., a healthy female volunteer, after ≥12 h fasting using a venous cannula. NH_4_-Heparin served as an anti-coagulant. The blood sample was centrifuged at 2500× *g* and 12 °C for 15 min. The obtained blood plasma was then allowed to pass through a 0.45 µm filter by gravity flow to eliminate cellular debris before EV isolation.

### 2.5. Isolation of EVs

For EV preparation from cell culture supernatants, cells were cultured for at least 48 h in DMEM supplemented with exosome-depleted FCS (Cat. No. A25904DG, Life Technologies, Carlsbad, CA, USA). Conditioned cell culture media (CM) were depleted of cellular debris by differential centrifugation for 10 min at 300× *g* and for 20 min at 2000× *g*. After concentrating the media to 1/10th of the original volumes by centrifuging at 2500× *g* and 12 °C in Vivaspin ultrafiltration units (Cat. No. VS2092, Sartorius, Göttingen, Germany), the respective concentrated CM (CCM) samples were used for EV isolation. EVs were either prepared by polymer-based precipitation (ExoQuick-TC, Cat. No. EXOTC10A-1, System Biosciences, Palo Alto, CA, USA) according to the manufacturer’s instructions or by differential centrifugation. For differential centrifugation, the respective CCM samples were subjected directly to ultracentrifugation (100,000× *g*, 4 °C, 90 min), twice, with a washing step in between (washing buffer: 100 µL PBS supplemented with protease inhibitor cocktail), if not stated otherwise. Both approaches led to an enrichment of diverse EVs including smaller ones, e.g., exosomes, and larger subtypes, e.g., MVs, since the latter ones were not depleted by a 10,000× *g* centrifugation step.

To obtain highly pure vesicles of smaller sizes, density gradient centrifugation was used. In that case, larger EVs were depleted by an upstream 10,000× *g* centrifugation step prior to ultracentrifugation. Next, the resulting EV pellet was resuspended in washing buffer and applied to an OptiPrep density gradient as described by Lobb et al. [37]. After refractometric determination of the respective densities, the obtained fractions were concentrated by ultracentrifugation as described above and the resulting EVs were analysed by immunoblotting.

### 2.6. Proximity Labelling

The APEX2-mediated proximity labelling of the cells and streptavidin-based pull-down of biotinylated proteins was conducted as described by Hung et al. [38]. For the biotinylation of EV proteins, two labelling strategies were tested (Figure 3). In strategy “a”, the reagents were directly added to the conditioned cell culture media with subsequent EV isolation. In strategy “b”, first, the EVs were isolated. The obtained EV pellet was then labelled and quenched like the cells, while great care had to be taken not to disturb the pellet. For both strategies “a” and “b”, EVs were prepared using polymer-based precipitation.

### 2.7. Immunoblotting (IB)

Cells and EVs were lysed in RIPA buffer according to Hung et al. [38]. Proteins were precipitated with methanol/chloroform, dried at room temperature (RT), and resuspended in 2% SDS. The protein concentration was determined using the *DC* Protein Assay (Cat. No. 5000116, Bio-Rad Laboratories, Hercules, CA, USA) according to the manufacturer’s instructions. The samples were incubated for 10 min at 45 °C in sample buffer (40% (*v*/*v*) glycerol, 8% (*v*/*v*) SDS, 225 mM tris-HCl (pH 6.8), 8% (*v*/*v*) β-mercaptoethanol, 5 g/L of bromophenol blue sodium salt) and subjected to SDS-PAGE (Any kD Mini-PROTEAN TGX Stain-Free Protein Gels, Cat. No. 4568124 or 4568126, Bio-Rad Laboratories). Per sample and gel lane, 25 µg of total protein were applied. Protein amounts (loading controls) were visualised according to the “stain-free” technique, which enables the direct visualisation of proteins in gels supplemented with 2,2,2-trichloroethanol (TCE) by an ultraviolet light-induced reaction of tryptophan residues with the trihalo compound that produces fluorescence in the visible range [39]. Following stain-free imaging, the proteins were transferred to a PVDF membrane using Trans-Blot Turbo Mini PVDF Transfer Packs (Cat. No. 1704156, Bio-Rad Laboratories). The membranes were blocked in an appropriate blocking agent (5% (*w*/*v*) milk powder in TBS-T (0.05% Tween-20 in TBS) or 3% bovine serum albumin (BSA) in TBS-T) at RT for 2 h, incubated with primary antibody overnight at 4 °C, washed in TBS-T, incubated with peroxidase-coupled secondary antibody for 1 h at RT, and washed again in TBS-T. Blots were visualised by chemiluminescence using Clarity Western ECL Substrate (Cat. No. 1705061, Bio-Rad Laboratories) and the ChemiDoc Imaging System (Bio-Rad Laboratories).

### 2.8. Mass Spectrometric Protein Identification and Quantification

Eluted protein samples were processed for mass spectrometric analysis by in-gel digestion, essentially as described [40]. Briefly, samples were separated (~5 mm running distance) in Bis-Tris buffered 4–12% acrylamide gels (Thermo Fisher Scientific, Waltham, MA, USA). After silver staining and de-staining of the gel, protein-containing bands were washed, reduced by DTT, alkylated by adding iodoacetamide, and digested with 0.1 µg trypsin in 50 mM ammonium hydrogen carbonate overnight at 37 °C. Peptides were extracted from the gel by twice adding 40 µL of 1:1 (*v*/*v*) acetonitrile and 0.1% trifluoroacetic acid (in water), followed by a 15 min incubation in an ultrasonic bath. Peptides were dried and finally resuspended in 17 µL of 0.1% trifluoroacetic acid.

Next, peptide samples were separated on an Ultimate 3000 rapid separation liquid chromatography system (RSLS, Thermo Fisher Scientific). First, peptides were trapped on a trapping column (Acclaim PepMap100, 3 µm C18 particle size, 100 Å pore size, 75 µm inner diameter, 2 cm length, Thermo Fisher Scientific) using 0.1% (*v*/*v*) trifluoroacetic acid (in water) as the mobile phase. After 10 min, the peptides were separated at 60 °C on an analytical column (Acclaim PepMapRSLC, 2 µm C18 particle size, 100 Å pore size, 75 µm inner diameter, 25 cm length, Thermo Fisher Scientific) for 2 h (gradient from 4% to 40% solvent (0.1% (*v*/*v*) formic acid, 84% (*v*/*v*) acetonitrile in water). The separated peptides were directly analysed by an online coupled QExactive plus quadrupole/orbitrap mass spectrometer (Thermo Fisher Scientific). Peptides were sprayed into the mass spectrometer via an online coupled nano source equipped with distally coated emitters (New Objective) at a spray voltage of 1.4 kV. The mass spectrometer was operated in data dependent positive mode: first survey scans were carried out at a resolution of 70,000 over a scan range of 350 to 2000 *m*/*z*. Subsequently, up to ten twofold- and threefold-charged precursors were isolated by the build in quadrupole (2 *m*/*z* isolation window) and fragmented via higher-energy collisional dissociation. Fragment spectra were recorded at a resolution of 17,500 using an available scan range of 200 *m*/*z* to 2000 *m*/*z*, and already-fragmented precursors were excluded from fragmentation for the next 100 s.

Raw data were further processed with MaxQuant (version 1.6.3.4) with standard parameters if not indicated otherwise, enabling protein identification and quantification. Searches were carried out with tryptic cleavage specificity considering two potential missed cleavage sites based on 73,112 *Homo sapiens* entries downloaded from UniProtKB (UP000005640) on 18th August 2018 and, additionally, the APEX2-GABARAP sequence. Carbamidomethylation at cysteines was considered as fixed and methionine oxidation as variable modification. Label-free quantification was enabled (LFQ). Only proteins identified with at least two peptides were reported.

For data analysis, normalised (LFQ) intensities were used to determine the enrichment of a protein in one sample type relative to another. Gene Ontology (GO) analyses were conducted by applying protein lists to DAVID Bioinformatics Resources 6.8 [41], using either the *Homo sapiens* database as the default background or Appendix A as the background of the total proteins detected in this study.

To generate separate proteins lists for each sample, raw data were processed using the ProteomeDiscoverer 1.4 software environment (Thermo Fisher Scientific). Searches were carried out using the MSAmanda search engine with tryptic cleavage specificity considering one potential missed cleavage site based on 42,201 *Homo sapiens* SwissProt entries downloaded from UniProtKB on 20th June 2017 and, additionally, the APEX2-GABARAP sequence. Carbamidomethylation at cysteines was considered as fixed, and deamidation at asparagine and glutamine as well as methionine oxidation, as variable modifications. Mass tolerances were 10 ppm both for precursors and fragments. Identifications were validated using the Percolator node and only “high confidence” peptides (FDR <1%) used for further processing. Only proteins identified with at least two peptides were reported.

### 2.9. Immunofluorescence (IF)

HEK293 cells (4 × 10^5^) were seeded into poly D-Lysine coated bottom dishes (Cat. No. P35GC-0-10-C, MatTek Corporation), incubated in DMEM/10% FCS, and transfected on the following day. After 48 h of further cultivation, the cells were incubated for 2 h at 37 °C in Hank’s Balanced Salt Solution (HBSS; Cat. No. 14025050, Thermo Fisher Scientific) and 100 nM Bafilomycin A1 (Cat. No. 196000, Sigma-Aldrich) to induce the accumulation of autophagic structures. Then, the cells were fixed at 37 °C for 10 min with 4% (*w*/*v*) paraformaldehyde (PFA) in PBS, pH 7.4; washed two times with PBS; and permeabilised by shaking in 0.2% Triton-X-100 for 30 min at RT. After another washing step, the cells were blocked by incubating in 1% BSA overnight at 4 °C. Immunostaining was conducted by adding primary anti-myc antibody, diluted in anti-GABARAP (8H5) antibody, and incubating for 60 min at RT under smooth shaking. Again, the cells were washed three times and then incubated with shaking for 60 min at RT in secondary antibodies under the exclusion of light. After two final washing steps, long storage buffer was added (0.05% sodium azide in PBS) and the cells were subjected to image acquisition.

### 2.10. Image Acquisition—Laser Scanning Microscopy (LSM)

Images were acquired using an LSM 710 confocal microscope (Zeiss, Oberkochen, Germany) equipped with the ZEN Black 2009 software and a Plan-Apochromat 63x/1.40 Oil DIC M27 objective. Cell nuclei (DAPI) were visualised using the 405 nm channel (MBS-405); GABARAP, using the 488 nm channel (MBS 690+); and myc, using the 543 nm channel (MBS 458/543).

### 2.11. Transmission Electron Microscopy (TEM)

EVs were prepared for TEM according to Théry et al. [42] with slight alterations. After enriching the EVs using polymer-based precipitation, they were resuspended in 2% (*w*/*v*) PFA in PBS, pH 7.4. Of this sample, 5 µL were absorbed on a formvar/carbon coated copper grid (Cat. No. S160-4-V, Plano, Wetzlar, Germany) for 20 min. The specimen was fixed with 1% (*v*/*v*) glutaraldehyde for 5 min, washed eight times with ddH_2_O, and negative stained with 4% (*v*/*v*) uranyl acetate for 5 min. Finally, the samples were embedded in a freshly made mixture (4:1) of trehalose (10%, in TBS) and uranyl acetate (2%) for 10 min at RT, and then dried. Images were recorded using a Libra 120 transmission electron microscope (Zeiss, Oberkochen, Germany) at 120 kV.

### 2.12. Nanoparticle Tracking Analysis (NTA)

HEK293 cells were cultured for 48 h in phenol red-free DMEM (Cat. No. 21063029, Thermo Fisher Scientific) supplemented with exosome-depleted FCS (Cat. No. A25904DG, Life Technologies). Conditioned cell culture media were depleted of cellular debris by differential centrifugation for 10 min at 300× *g* and for 20 min at 2000× *g*. The media were concentrated to 1/10th of the original volumes by centrifuging at 2500× *g* and 12 °C in Vivaspin ultrafiltration units (Cat. No. VS2092, Sartorius), stored overnight at 4 °C, and measured by NTA the day after. Measurement of the samples was conducted with a ZetaView PMX-100 (Particle Metrix, Meerbusch, Germany) using a 1/10 dilution in Ampuwa water (Cat. No. 1088813, Fresenius Kabi, Bad Homburg, Germany). Additionally, the particle size distribution in diluted non-conditioned medium was determined and used for background subtraction. The following acquisition parameters were used: Sensitivity—75%; Shutter—70; Minimum Brightness—20; Minimum Size—20 nm; Maximum Size—500 nm; Polarity—1; Voltage—0; Particle drift at 0 V—<5 µm/s; Positions—11; Cycles—10; and Multiple acquisitions—2. Each sample was measured in two independent trackings, which were used for the calculation of standard deviations. The measured particle counts per size were corrected by background subtraction.

## 3. Results

### 3.1. GABARAP is Secreted in EVs of Different Cell Lines and Sample Types

First, we checked whether GABARAP is secreted in EVs by those cell lines commonly used in cell culture experiments. To this end, we analysed cell culture supernatants from the human cell lines HEK293, Huh7, and SH-SY5Y after inducing the accumulation of autophagic structures by starvation and using an anti-GABARAP antibody, the GABARAP-specificity of which we confirmed before by knockout validation (Appendix A in [36]). For all three cell types, GABARAP was detected in the respective EV sample (Figure 1A). Notably, predominantly the lipidated form of GABARAP, GABARAP-II, was found, according to its migration behaviour [43]. An enrichment of EVs in the respective samples was confirmed by transmission electron microscopy (TEM) and also by nanoparticle tracking analysis (NTA; Appendix A), which resulted in particle sizes ranging from approximately 30 to 1000 nm (peak at 200 nm) as expected for EVs derived by differential centrifugation with an omitted run at 10,000× *g*. In addition, we applied EVs isolated both from HEK293 cells and from HEK293 cells stably overexpressing HIV-1 Nef—a protein that is well-known to trigger both its own and EV’s secretion in general [44]—to density gradient centrifugation and subsequent immunoblotting. We found that GABARAP co-fractionated with Alix as a well-defined EV marker but also with the Nef protein, when present, at a buoyant density characteristic for exosomes and other EVs [45]. In both cases, GABARAP was not found in fractions of higher density, hinting towards a localisation of GABARAP in EVs rather than in protein aggregate contaminants (Appendix A). Next, we wer e curious to see whether GABARAP was also secreted in EVs from “fed” HEK293 cells without inducing the accumulation of autophagic structures and containing predominantly unlipidated GABARAP-I. Interestingly, in the respective EV-enriched samples obtained from CCM of cells cultured under nutrient-rich conditions, we again detected mainly lipidated GABARAP-II (Figure 1B, EVs), suggesting its association with the vesicle membrane. As control, we also analysed unconditioned media (UCM), which were treated just as conditioned media. Unexpectedly, for UCM, a faint signal for unlipidated GABARAP-I was detected, which might have its origin in FCS components remaining in the applied FCS even under exosome depletion. Note that the human and bovine GABARAP and Alix proteins are identical by 100% and 95%, respectively. Consistently, in EVs prepared from FCS without prior exosome depletion, GABARAP-I was again observed (Figure 1C). The reason why in FCS-derived EVs, GABARAP is found in its unlipidated form is unknown; however, this fact supports the idea that GABARAP-II originates from those EVs produced by the cultured cells in all our CCM-derived EV preparations. Finally, we detected GABARAP-II also in EVs isolated from human blood plasma samples (Figure 1D). Altogether, these results demonstrate that secretion of endogenous GABARAP is widespread.

### 3.2. APEX2-GABARAP Exhibits Different Cellular Localisation Than APEX2’s

To get an idea about the distribution of APEX2-tagged GABARAP, cells overexpressing APEX2-GABARAP or APEX2 alone were starved and imaged by confocal laser-scanning microscopy, and their intracellular localisation was compared with that of endogenous GABARAP (Figure 2A). As expected, a broad co-localisation pattern between the anti-myc stain, corresponding to APEX2-GABARAP due to its N-terminal myc tag, and the highly GABARAP-specific antibody 8H5 [36], which detects both endogenous GABARAP and its APEX2-fusion, was observed. By contrast, cells expressing APEX2 alone showed virtually no overlap between the two stains. In agreement with the culture conditions used, APEX2-GABARAP was prominently found in punctate structures, very likely resembling its lipidated and thus autophagosome-associated fraction. For APEX2 alone, a uniform distribution throughout the cytoplasm was observed as expected for a soluble protein.

Next, we confirmed the peroxidase activity of APEX2-GABARAP by adding biotin-phenol and H_2_O_2_ to the cell cultures, following the protocol as published by Hung et al. [38] and subsequent immunoblotting of the respective cell lysates using a Streptavidin-HRP conjugate. Different biotinylation patterns (Figure 2B) were observed in accordance with the differing intracellular distribution of APEX2 and APEX2-GABARAP. In consistency with its vesicular localisation, a rather distinct biotinylation pattern and thus a restricted pool of proximate proteins was obtained for APEX2-GABARAP, while for APEX2 alone, a very broad biotinylation pattern was observed, reflecting its high mobility and wide variety of putative interacting partners. Furthermore, we prepared EVs from APEX2-GABARAP-expressing cells and confirmed the secretion of overexpressed APEX2-GABARAP (Figure 2D), being consistent with the secretion of endogenous GABARAP (Figure 1). Related to this, by investigating an EGFP knock-in cell line expressing EGFP-GABARAP under the control of the endogenous GABARAP promotor (Appendix A, [46]), we could show that EGFP-GABARAP can also be robustly detected by immunoblotting in the respective EV lysates yielded after ultracentrifugation when present at endogenous levels. This indirectly suggests that overexpression is, most likely, not the (sole) explanation for the observed APEX2-GABARAP secretion, which instead should rather reflect the secretion behaviour of GABARAP as documented in Figure 1. The enrichment of EVs from APEX2-GABARAP expressing cells was finally assessed by TEM (Figure 2C) and the immunoblotting of EV lysates against two well-accepted EV marker proteins (Alix, CD81) and calnexin to exclude cellular contaminants (Figure 2D), respectively.

### 3.3. APEX2-GABARAP Proximity Labelling in EVs

Having demonstrated the functionality of APEX2-GABARAP and also its export in EVs, we subjected EV-enriched samples, containing both EVs positive for APEX2-GABARAP and EVs without APEX2-GABARAP, to proximity labelling followed by streptavidin-pulldown and MS analysis with the aim of defining the GABARAP co-secretome. Since we could detect endogenous GABARAP in EVs both under starvation as well as under basal conditions (Figure 1A,B) and also overexpressed APEX2-GABARAP in EVs prepared under basal conditions (Figure 2D), we cultured the cells without nutrient deprivation to minimise stress-related artefacts. To establish an efficient labelling result, we tested two different strategies, for both of which we proceeded according to the protocol published by Hung et al. [38]. In one approach (simultaneous labelling, Figure 3A, branch a), we added the labelling reagents directly to the conditioned media after cultivating HEK293 cells for 48 h, thereby labelling cellular and EV proteins at the same time and then separating them for analysis. In the other approach (separated labelling, Figure 3A, branch b), cells and EVs were separated from each other in the first step and labelled in the second step. For the EVs, this included an upstream preparation step. The pelleted and concentrated EVs were then labelled in the same procedure as the cells. The total cell lysates and EVs of both approaches were then analysed for biotinylation by immunoblotting using a streptavidin–HRP conjugate. While for the cellular sample, biotinylated proteins were detected in both the cases tested, EV content biotinylation was visible only for EVs that have been labelled separately subsequent to their isolation (Figure 3B). Consistently, under these conditions, the affinity-purification of biotinylated proteins by streptavidin-coated beads could be demonstrated (Figure 3C).

### 3.4. The GABARAP Co-Secretome as Defined by Proximity Labelling

Finally, we were curious about the identity of those proteins detected in the EVs from APEX2-GABARAP expressing cells after labelling and streptavidin-pulldown, since they define the co-secretome of GABARAP. For this, we cultured the cells under nutrient-rich conditions to obtain the co-secretome of APEX2-GABARAP under basal conditions and subjected the cellular and EV proteins, both labelled and non-labelled, to mass spectrometry (MS) and subsequent analysis using FunRich [47,48]. When examining the cytoplasmic samples (Figure 4A), it becomes apparent that the number of identified proteins in the cellular GABARAP proxitome (total 347) after labelling and streptavidin-pulldown is 64.7% lower than the number identified in the total cellular lysate from untreated control (ctrl) cells (total 983 proteins). Of the cellular proteins, a total of 212 proteins corresponding to 61.1% of the cellular GABARAP proxitome was also identified without labelling and pulldown, and 135 proteins or roughly one-third were detected exclusively after proximity-labelling. While the latter might include false-positive hits, the former consists of more than 50% of the proteins also found in the autophagosomal GABARAP proxitome [8] (Appendix A) and therefore probably represents true-positive hits. Altogether, this result met the expectation of a smaller, possibly more specific, number of proteins after labelling and pulldown.

In case of the EV samples (Figure 4B), a slightly elevated number of proteins (584) could be identified from the proximity-labelled sample when compared to the untreated control sample (EV ctrl proteome, 443). This could reflect either a technical-based variance between the samples or, e.g., reflect the enrichment of a multitude of little-abundant proteins by the proximity-labelling procedure raising them just beyond the MS detection limit. Another explanation could be that APEX2-GABARAP shows a ubiquitous distribution through different EV subtypes, which would be consistent with the broad list of identified proteins observed. This could lead to biotinylation and the subsequent detection of proteins that are localised together with APEX2-GABARAP in EVs but are not necessarily in proximity to intracellular APEX2-GABARAP.

In general, the EV GABARAP proxitome contained various proteins commonly found in EVs, like 14-3-3 proteins (YWHAB, YWHAE, YWHAQ, YWHAZ), annexin proteins (ANXA2P2 and ANXA5), cytoskeletal proteins (ACTB, ACTR3, CFL1, PFN1, TUBA1B, TUBA1C, TUBB, TUBB3, and TUBB4B), heat shock proteins (HSP90AA1, HSP90AB1, HSPA6, HSPA8, and HSPD1), metabolic proteins (AHCY, ENO1, FASN, GAPDH, GSTP1, PGK1, PPIA, and UGDH), motor proteins (DYNC1H1, DYNC1I2, MYH10, and MYH9), RNA-binding proteins (RBPs) (e.g., HNRNPK, HNRNPM, HNRNPR, and HNRNPU), small GTPases (ARF3, IQGAP1, RAB1A, RAB1B, RAB10, RAB14, RAB35, RAB7A, RAN, RANGAP1, RANBP1, and RANBP2), transporters and channel proteins (ATP1A1, ATP5B, SLC25A3, SLC25A5, SLC25A6, SLC25A13, VDAC1, VDAC2, and VDAC3), and vesicle trafficking-associated proteins (Alix and VCP). A total of 216 proteins (40.0%) of the EV GABARAP proxitome was also detected in the unlabelled EV control proteome. By calculating the quotient of normalised LFQ intensities, we determined 68 proteins of the EV GABARAP proxitome enriched by a factor of > 2 compared to in the EV ctrl proteome (Appendix A). Among these enriched proteins, outstanding examples are the sodium/potassium-transporting ATPase subunit alpha-1 ATP1A1, which is required for autophagy induction [49] and is contained in clathrin-coated vesicles and earlier endosomes [50], suggesting an involvement in trafficking processes; Rab GDP dissociation inhibitor beta (GDI2), which was shown to regulate exosome maturation and secretion [51]; L-lactate dehydrogenase B chain (LDHB), a protein taking part in vesicle maturation and necessary for basal autophagy [52]; peroxiredoxin-1 (PRDX1), being released in exosomes [53]; and ubiquitin-like modifier-activating enzyme 1 (UBA1), which is required for Atg7- and Atg3-independent autophagy [54]. Additionally, these proteins are registered in Vesiclepedia [28,29] in the list of the top 100 proteins previously found in EVs. In total, of the EV GABARAP proxitome, more than 95% is registered in Vesiclepedia (Appendix A), and 63 proteins were found in the Vesiclepedia top 100 (Appendix A).

Next, we compared the protein lists obtained from the cellular and EV GABARAP proxitomes (Figure 4C) and noticed that 53.0% of the proteins identified in the cellular GABARAP proxitome were also included in the EV GABARAP proxitome. In addition, when correlating the EV GABARAP proxitome with the autophagosomal GABARAP proxitome as determined by Le Guerroué et al. [8], we found that of the EV GABARAP proxitome, 64.4% (376 proteins) was detected before as proteins in proximity to autophagosomal APEX2-GABARAP (Figure 4D). We also compared the EV GABARAP proxitome with the exosomal proteome of 293T cells as determined by Li et al. [55], showing an overlap of 84.1% (491 proteins) (Figure 4E).

The complete protein hit lists of the EV GABARAP proxitome (Figure 5A,B), the cellular GABARAP proxitome (Figure 5C), the unlabelled cellular ctrl proteome (Figure 5D), the unlabelled EV ctrl proteome (Figure 5E), the 293T exosome proteome [55] (Figure 5F), and the autophagosomal GABARAP proxitome [8] (Figure 5G) were then subjected to an in-depth Gene Ontology (GO) analysis using DAVID Bioinformatics Resources 6.8 [41].

As expected, in the GO cellular compartment analysis of the EV GABARAP proxitome (Figure 5A), we found a high percentage of the detected proteins to be assigned to extracellular vesicles (53.5%), while this category was less represented in the cellular GABARAP proxitome (Figure 5C, 36.1%) and in the cellular ctrl proteome (Figure 5D, 45.9%). Furthermore, of all here-analysed protein lists, in the EV GABARAP proxitome the highest percentage of proteins was assigned to focal adhesions (15.3%), in accordance with cell adhesion-related proteins being loaded into EVs [56]. Furthermore, we found, rather unexpectedly, a high percentage of nucleus-assigned proteins in the EV GABARAP proxitome (64.7%). Proteins assigned to extracellular vesicles and focal adhesions also showed a significant overrepresentation when the complete list of proteins identified in this study was set as background; however, this was not the case for nucleus-assigned proteins (Appendix A). An additional GO analysis of the EV GABARAP proxitome revealed RBPs as the top term regarding molecular functions (Figure 5B).

Intriguingly, we also detected 23.7% of the EV GABARAP proxitome to be listed and significantly overrepresented as mitochondrion-annotated proteins. Again, mitochondrion-assigned proteins also showed significant overrepresentation against the background protein list of this study (Appendix A). By contrast, for both the whole EV ctrl proteome (Figure 5E) and the independent 293T exosome proteome [55] (Figure 5F) less than 10% of the proteins were mitochondrion-annotated and did not show significant enrichment. Interestingly, in the autophagosomal GABARAP proxitome [8] (Figure 5G), 34.2% of the proteins were assigned to mitochondria and significantly enriched in this compartment. With mitochondria being a well-defined cargo of autophagosomes [8], comparative GO analysis of the different data sets therefore hints towards an autophagosomal origin for at least a fraction of those EVs that are positive for (APEX2-)GABARAP.

Next, we analysed the proteins enriched >2-fold in the EV GABARAP proxitome compared to the EV ctrl proteome (Appendix A). Of the 68 proteins, 56 proteins were also identified in the autophagosomal GABARAP proxitome [8], of which 27 proteins were assigned to the mitochondrion using FunRich [48]. An interesting example for this category is 60 kDa heat shock protein (HSPD1), which is not only mitochondrial but is also trafficked to the cell surface and is released into the extracellular space [57]. Another heat shock protein exhibiting a high affinity for phospholipid membranes, HSP90B1 [58], was identified. Complementary to the mitochondrial proteins, we found the autophagy-involved proteins cofilin-1 (CFL1) [59], interleukin enhancer-binding factor 3 (ILF3) [60], phosphoglycerate kinase 1 (PGK1) [61], poly(rC)-binding protein 1 (PCBP1) [62], and transcription intermediary factor 1-beta (TRIM28) [63]. A role in the modulation of autophagy was also revealed for the heterogeneous nuclear ribonucleoprotein K (HNRNPK) and M (HNRNPM) [61,64]. Further ribosomal proteins—namely RPL5, RPL6, RPL12, RPL18, RPL22, RPS16, and RPS4X—were identified, possibly due to their RNA-binding property or, as recently reported, being present on the surface of exosomes [65].

Furthermore, we had a closer look at those proteins that were discovered specifically within the EV GABARAP proxitome, but neither in the EV ctrl proteome nor in the 293T exosome proteome, and set an intensity threshold of >24, corresponding to 75% of the highest normalised intensity. Of the 36 proteins meeting these requirements, 24 proteins were also found in the autophagosomal GABARAP proxitome, of which, again, many (20 proteins) are assigned to mitochondria by FunRich [48]. Examples of mitochondrial enzymes are the MICOS complex subunit MIC60 (IMMT), solute carrier family 25 member 13 (SLC25A13), and cytochrome b-c1 complex subunit 1 (UQCRC1). IMMT was previously demonstrated to be targeted by LC3C, a human Atg8 orthologue like GABARAP, to deliver mitochondrial proteins to the autophagosome [8]. Furthermore, we detected the acetyl-CoA acetyltransferase (ACAT1) and 7-dehydrocholesterol reductase (DHCR7) lipid metabolism-related proteins, and the trafficking proteins Rab7A (late endosome-to-lysosome traffic), Rab10 (exocytosis), and Rab14 (recycling endosome-to-plasma membrane (PM) traffic (Rab functions reviewed in [66]).

Next, we were curious to see whether proteins found in the intersection of the autophagosome GABARAP proxitome and the EV GABARAP proxitome (Figure 5D) contain an LC3-interacting region-motif (LIR), which can frequently be found in proteins directly interacting with both the LC3s and the GABARAPs of the Atg8 protein family [67]. For this, we applied a sequence scan for an extended LIR motif (xLIR) [68] using the web-based ScanProsite tool [69]. Of the 376 applied proteins, we found 271 motif hits in 164 sequences (Appendix A). These hits were compared with the IntAct database, revealing 60 proteins as registered GABARAP interactors (https://www.ebi.ac.uk/intact/interactors/id:O95166*, accessed on 5 July 2019), for example transferrin receptor protein 1 (TFRC) [25], calreticulin (CALR) [70], and cathrin heavy chain 1 (CLTC) [71]. Furthermore, we identified coatomer subunit delta (COPD) as a trafficking protein and secretion-related proteins like Alix; kinesin-1 heavy chain (KIF5B), which is a protein involved in GABARAP-mediated transport [27]; and La-related protein 1 (LARP1), which was shown to take part in autophagy [72]. Future work should include a detailed study on the exact location, accessibility, and context of each potential GABARAP-binding motif identified. Such work is indispensable for each hit to better estimate whether direct binding to GABARAP is just random-based or actually feasible for the respective hit and thus potentially biological meaningful.

## 4. Discussion

The biological background behind GABARAP’s secretion, demonstrated in this study for diverse samples, is unknown. Several scenarios are conceivable including microvesicle shedding or secretion in exosomes via multivesicular bodies or an autophagy-mediated unconventional protein secretion process, so-called autosecretion, leading to the export of autophagosomal content like mitochondria rather than degradation [73]. However, it is unclear whether GABARAP is just secreted as a by-product of the trafficking and secretion of other molecules such as insulin in a cell type-specific manner [27] or whether there exists a not-yet-described GABARAP-involving branch of EV biogenesis and transport. Here, we provide information about the co-secretome of GABARAP in EVs as a basis to unravel GABARAP’s secretion mechanism.

Many studies have been conducted to analyse the protein content of EVs by MS (for examples, see [40,41]). Although it is possible to isolate specific EV fractions, e.g., by differential centrifugation, no method had been described to date to identify those proteins contained in the same EV as the POI, in this case GABARAP, and which thus define its co-secretome, which could be helpful in deciphering trafficking and secretion pathways. In principle, APEX2-proximity labelling [35] should be able to overcome this issue as this technique introduces biotin-modifications to those proteins that are localised in a 20 nm distance range surrounding the APEX2-POI-fusion. Because of the restricted sizes of EVs and particularly of exosomes, one can expect that a bulk of the protein content of each APEX2-POI-positive EV will be biotinylated and thus be marked for subsequent streptavidin-pulldown-based enrichment and identification via MS. In this regard, the most challenging task when adapting the APEX2-strategy to EV samples was coping with the small size of the particles, 40–100 nm (exosomes) and 100–1000 nm (microvesicles) [39], which were present in highly diluted form in the conditioned cell culture media. In this study, we were able to achieve the labelling of EV proteins with one of the two approaches tested. We therefore suggest an upstream concentration step for EVs when applying APEX2-mediated proximity labelling to EVs to produce the efficient biotinylation of proteins within APEX2-POI-positive EVs.

In defining the EV GABARAP proxitome, we were able to determine the first parameters in GABARAP’s secretion. Proteins taking part in vesicle-mediated transport processes were found, like COPD, as were proteins involved in exosomal secretion such as Alix, GDI2, and PRDX1. Furthermore, we detected known interaction partners of GABARAP such as CALR and CLTC [70,71], and also TFRC and KIF5B, which are involved in PM-directed transport processes together with GABARAP [25,27], possibly enabling the co-secretion of the latter. Comparing the EV GABARAP proxitome with the autophagosomal GABARAP proxitome as defined by Le Guerroué et al. [8], almost two-thirds of the EV GABARAP proxitome was also detected in autophagosomes. An in-depth analysis of proteins enriched in the EV GABARAP proxitome compared to in the EV ctrl proteome and an independent exosome proteome [55] revealed a high percentage of mitochondrion-assigned proteins in the EV GABARAP proxitome, pointing towards a possible (auto-)secretion mechanism for autophagosomal content specifically in EVs containing APEX2-GABARAP, but not in the bulk of EVs. Additionally, one could also speculate about an amphisome-mediated secretion pathway. Here, autophagosomes may fuse with multivesicular bodies and subsequently release their content into the extracellular space following fusion with the PM [74].

The use of a polymer-based EV isolation method prior to the APEX2-labelling step could be deemed as one limitation of this study. However, we chose this EV isolation technique due to its broad applicability and its ability to produce high yields of EVs regardless of their subtype, e.g., without excluding larger ones like microvesicles. That way we might have co-isolated contaminants such as protein aggregates to some extent [75], but this seems to be acceptable in our eyes, because the vast majority of them should be inaccessible to APEX2-mediated biotinylation and the following streptavidin-based pulldown in the given setup. However, we cannot completely exclude the observation of some “false-positives” within our GABARAP proxitome-derived MS data set due to unspecific co-isolates instead of biotinylated cargo of APEX2-GABARAP positive EVs. Electron microscopic measurements of EVs following the immunogold staining of GABARAP or 3,3′-diaminobenzidine (DAB) staining of APEX2-GABARAP might provide clarity in this respect. Nevertheless, shortly before the submission of our study, an approach based on BirA*-mediated proximity labelling was successfully applied to identify targets of LC3-dependent secretion. This study is interesting, because LC3, like GABARAP, belongs to the highly homologous and partly functionally redundant mammalian ATG8 protein family [21]. By labelling proteins in proximity to BirA*-LC3 intracellularly prior to secretion, Leidal et al. showed that parts of the autophagic machinery, more precisely the LC3-conjugation machinery, are required for the secretion and packaging of diverse RBPs in EVs that are enriched in lipidated LC3 [76]. When comparing our EV GABARAP proxitome, the “co-secretome” of APEX2-GABARAP, with the list of biotinylated proteins significantly enriched in EVs after intracellular BirA*-LC3 tagging, 38.7% of the BirA*-LC3 targeted proteins could also be identified in the EV GABARAP proxitome (Appendix A). Among them, as in the Leidal study, we identified the RBPs G3BP1, HNRNPK, LARP1, and SF3A1, which have already been demonstrated to associate with ATG8 family members in pulldown experiments. In this context, a new loading mechanism for RBPs into intraluminal vesicles of multivesicular bodies (MVBs) that relies on the LC3 conjugation machinery, neutral sphingomyelinase 2 (nSMase2), and an LC3-dependent recruitment of its regulator (factor associated with nSMase2 activity (FAN)) has been proposed [76]. Future studies are needed to clarify a putative functional redundancy of both GABARAP and LC3 regarding the underlying molecular mechanisms of “LC3-dependent EV loading and secretion”, a newly defined secretory pathway distinct from classical autophagy [76].

Taken together, this study provides the first information about proteins secreted together with GABARAP in EVs, indicating two pathways as possible mechanisms, including a putative autophagosome-mediated mechanism for, e.g., mitochondrial contents as well as an MVB-mediated pathway involving the ATG8-conjugation machinery for RBPs. In a broader context, we show that APEX2-mediated proximity labelling is applicable to secreted and isolated EVs from HEK293 cell culture supernatants and therefore might be a basis for disclosing the secrets of further biologically or medically relevant proteins contained in EVs, e.g., markers for cancer progression or diagnostic markers relevant for diverse neurodegenerative diseases.

## Figures and Tables

**Figure 1 cells-09-01468-f001:**
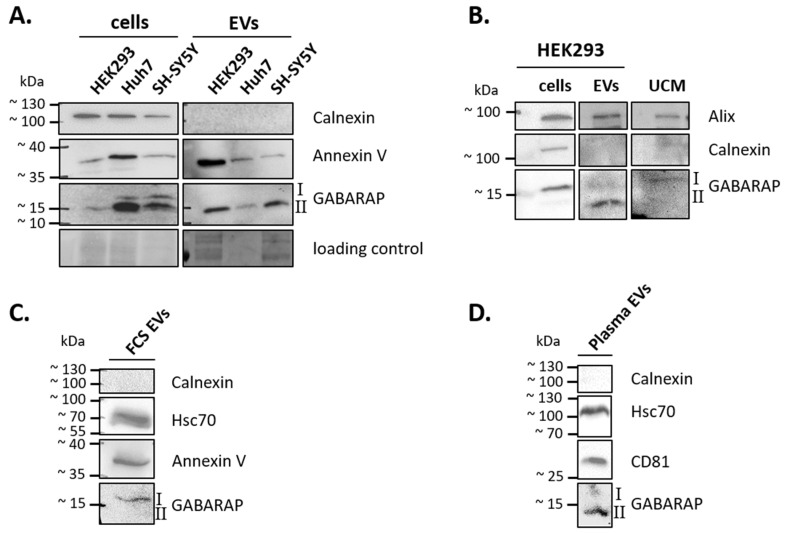
GABARAP is secreted in extracellular vesicles (EVs) and is detectable by immunoblotting. Alix, Annexin V, Hsc70, and CD81 were used as EV marker proteins, and Calnexin, as a marker for cellular impurities in an EV sample. (**A**) GABARAP is detected in EV-enriched samples from the three human cell lines—HEK293, Huh7, and SH-SY5Y—after starvation. One representative blot out of two, each performed with cell lysates from different passage numbers, is shown. EVs were obtained by ultracentrifugation. Cropped images of stain-free gels are presented as loading controls. (**B**) Under fed conditions, predominantly lipidated GABARAP is detectable in HEK293 EVs, while unlipidated GABARAP was detected in unconditioned media (UCM). One representative blot out of two, each performed with cell lysates from different passage numbers, is shown. EVs were enriched using polymer-based precipitation. (**C**) Unlipidated GABARAP is detectable in EVs isolated from bovine blood serum. One representative blot of two analysed EV batches, independently obtained from the same batch of FCS is shown. EVs were enriched by ultracentrifugation. (**D**) GABARAP is detectable in EVs isolated from human blood plasma. One representative blot of two analysed EV batches is shown. Respective EV samples were obtained from independent blood samples of the same donor. EVs were enriched by ultracentrifugation. Cells used in (**A**–**C**) were cultured for at least 48 h in exosome-depleted media before harvest. After lysis, cellular and EV proteins were precipitated by methanol–chloroform precipitation, resuspended in lysis buffer, and used for immunoblotting. Uncropped versions of the blots and their corresponding gels after stain-free imaging are given in Appendix A.

**Figure 2 cells-09-01468-f002:**
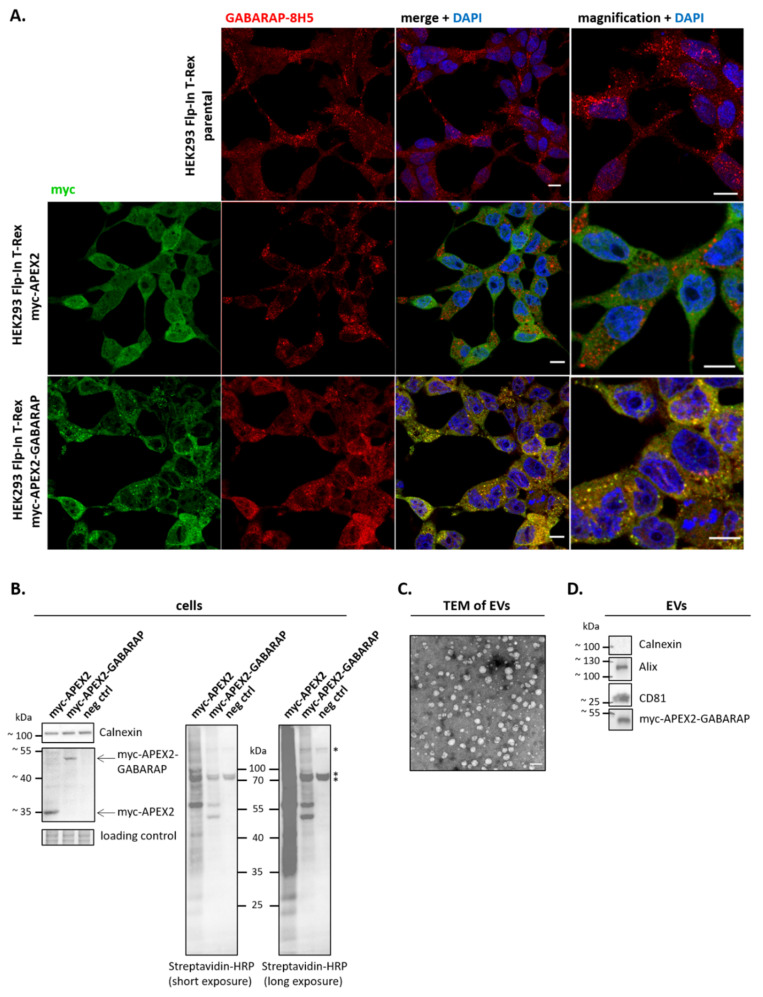
Cellular distribution of APEX2 and APEX2-GABARAP. (**A**) APEX2-GABARAP co-localises with the staining pattern obtained by the paralogue-specific anti-GABARAP (8H5) antibody [36] and shows a distribution pattern as observed for endogenous GABARAP. After a period of 24 h after transfection with APEX2 and APEX2-GABARAP, respectively, the cells were starved by incubating in HBSS + 100 nM BafA1 for 2 h at 37 °C and 5% CO_2_, fixed (4% paraformaldehyde, PFA), immunolabelled with goat anti-human c-myc antibody and anti-GABARAP-8H5, and visualised by confocal fluorescence microscopy. Parental, non-transfected cells are shown as control for the distribution of endogenous GABARAP. Nuclei were counterstained with DAPI. For each cell type, one representative image for three different cellular passages, each imaged in at least four frames, is shown. Scale bar, 10 µm. (**B**) Different biotinylation patterns are obtained when expressing APEX2-GABARAP compared to APEX2. HEK293 cells were transfected with APEX2- and APEX2-GABARAP-encoding plasmids, respectively. For the negative control (neg ctrl), no DNA was added. After an incubation period of 48 h, cells were labelled, harvested, and lysed as described by Hung et al. [38]. Cellular lysates were subsequently analysed by immunoblotting using a streptavidin–HRP conjugate. *: Endogenously biotinylated proteins. One representative blot out of four, each performed with cell lysates from different passage numbers, is shown. A crop of a stain-free gel is presented as the loading control. (**C**) Enrichment of EVs was tested by TEM. After preparation, EVs were resuspended in 2% PFA, contrasted in 4% uranyl acetate, embedded in 10% trehalose on formvar/carbon coated grids, and visualised by TEM. One representative image for three different cellular passages, each imaged in at least five frames, is shown. Scale bar, 100 nm. (**D**) EVs were enriched from cell culture supernatant by polymer-based precipitation after 48 h of cultivation and subjected to immunoblotting. Both the absence of calnexin as a cellular marker protein and the presence of common EV marker proteins (Alix, CD81) and of APEX2-GABARAP itself were confirmed. One representative blot out of three, each performed with cell lysates from different passage numbers, is shown. Uncropped versions of the blots and their corresponding gels after stain-free imaging are given in Appendix A.

**Figure 3 cells-09-01468-f003:**
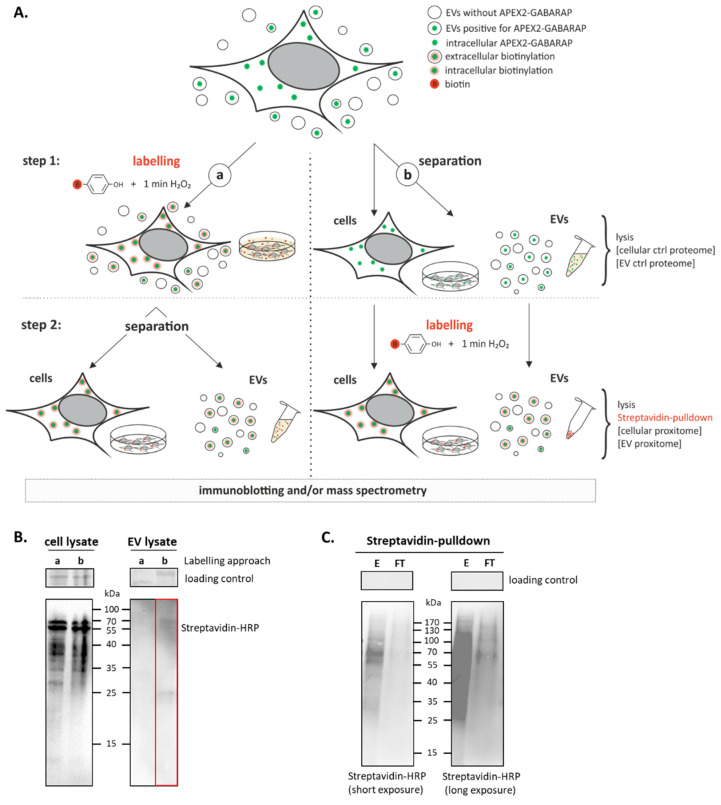
APEX2-mediated proximity labelling in EVs. (**A**) Tested approaches for proximity labelling of EVs. Two labelling strategies for target protein biotinylation within the APEX2-GABARAP-positive EV subfractions were investigated for their efficiency. In branch a, biotin-phenol and H_2_O_2_ were directly added to the cell culture supernatant in which the cells were cultivated for 48 h after transfection. Afterwards, the cells and EVs were separated from each other by collecting the supernatant. In branch b, cells and EVs were separated in the first step. EVs were pelleted by polymer-based precipitation. Then, cells and EVs were labelled separately. The cells and EVs were lysed and the labelled proteins can be analysed by immunoblotting using a streptavidin-HRP conjugate and/or mass spectrometry. (**B**) Pelleted EVs, but not EVs in solution, may be labelled by APEX2-GABARAP. The labelled EVs were lysed and used for immunoblotting, proving approach “b” suitable for APEX2-mediated labelling in EVs (marked red). One representative blot out of three, each performed with cell lysates from different passage numbers, is shown. (**C**) Biotinylated APEX2-GABARAP EV proteins, obtained using branch b, are enriched by streptavidin. EV lysates were incubated with streptavidin-coated beads. After collecting the flow-through (FT), multiple washing steps were applied, and the eluate (E) was collected. A broad signal was obtained for the eluate, while for the flow-through, only faint signals were obtained, demonstrating the effective capturing of biotinylated EV proteins. One representative blot out of two, each performed with cell lysates from different passage numbers, is shown. Crops of stain-free gels are presented as loading controls. Uncropped versions of the blots and their corresponding gels after stain-free imaging are given in Appendix A.

**Figure 4 cells-09-01468-f004:**
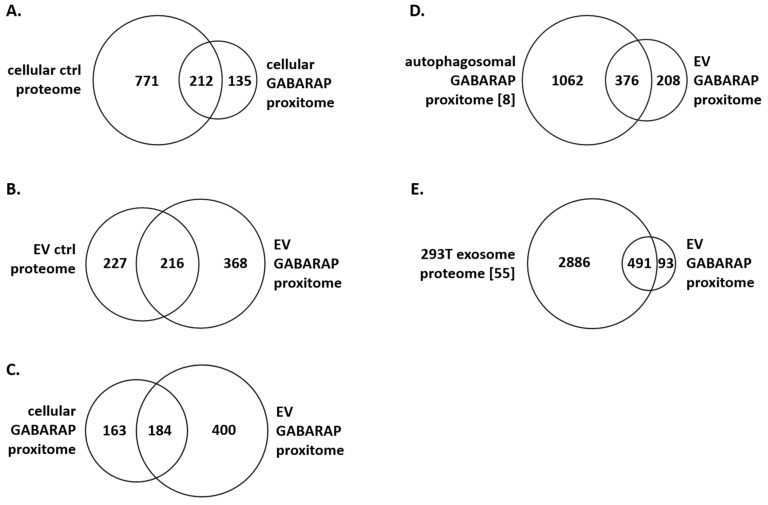
Overview of analysed sample types. Following APEX2-GABARAP expression, one sample type was collected without labelling (cellular (**A**) and EV (**B**) control (ctrl) proteome), while for the other sample type, cells and EVs were labelled and applied to pull-down with streptavidin (cellular (**A**) and EV (**B**) GABARAP proxitome). All samples were lysed and subjected to mass spectrometry. Cellular and EV GABARAP proxitomes were compared as shown in (**C**). The EV GABARAP proxitome was furthermore compared to the autophagosomal GABARAP proxitome as determined by Le Guerroué et al. [8] (**D**) and to the exosomal proteome of 293T cells as determined by Li et al. [55] (**E**). The Venn diagrams were created using FunRich [47,48].

**Figure 5 cells-09-01468-f005:**
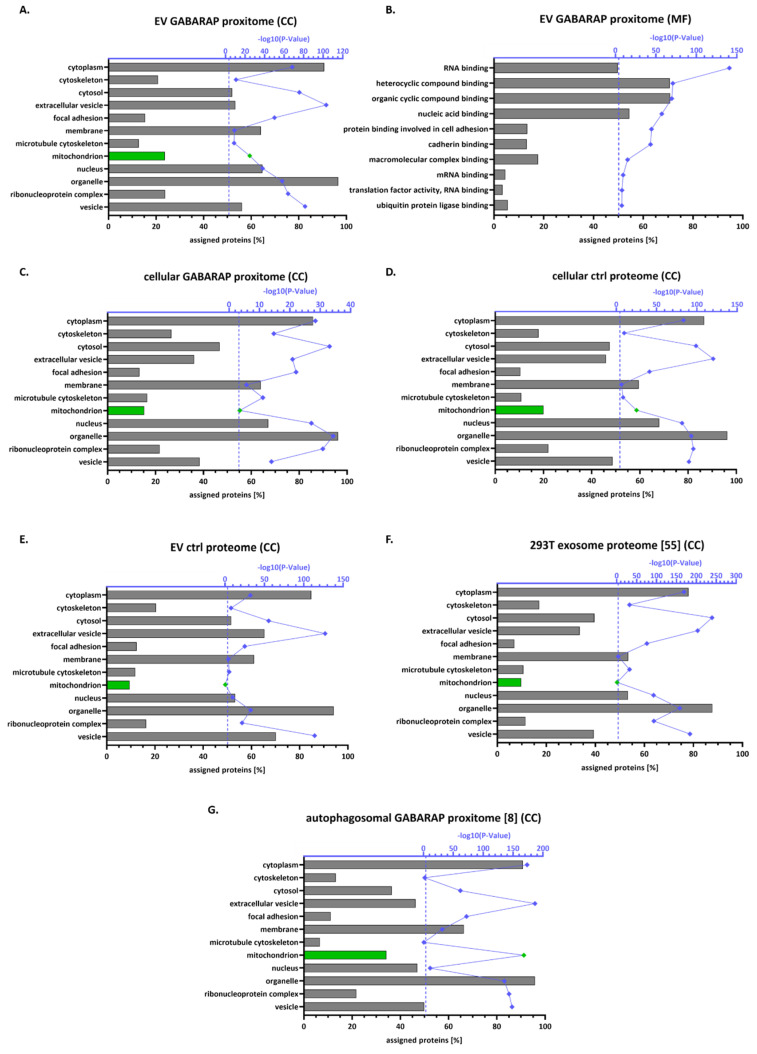
Categorisation of cellular and EV proteins detected by mass spectrometry. The obtained proteins for the EV GABARAP proxitome (**A**,**B**), the cellular GABARAP proxitome (**C**), the cellular ctrl proteome (**D**), the EV ctrl proteome (**E**), the 293T exosome proteome [55] (**F**), and the autophagosomal GABARAP proxitome [8] (**G**) were analysed by Gene Ontology (GO) using DAVID Bioinformatics Resources 6.8 (default background) [41] (CC: cellular compartment; MF: molecular function). The ratio of assigned proteins to the total protein number of the respective sample type is depicted in [%]. In blue, the negative log10 of the *p*-value is shown. Significantly overrepresented cellular compartments exhibiting a *p*-value of ≤0.05% (dashed line) in the EV GABARAP proxitome were selected. Mitochondrion-related bars [%] and -log10 (*p*-Value) data points are highlighted in green. For the complete data, see Appendix A. A comparative GO CC analysis of the EV GABARAP proxitome using Appendix A as background is depicted in Appendix A.

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
