# Peer review of "The GABARAP Co-Secretome Identified by APEX2-GABARAP Proximity Labelling of Extracellular Vesicles"

_cells, 2020, doi:10.3390/cells9061468_

Round 1

Reviewer 1 Report

In the manuscript titled: “the GABARAP co-secretome indentifed by APEX2GABARP proximity labelling of extracellular vesicles” by Sanwald et al, the authors show that GABARAP is expressed in or on EV. Furthermore they give an overview of the co-secretome of proteins secreted with GABARAP.

In general I have the feeling the data shown in the manuscript is insufficient. Without being disrespectful, but it feels like a ‘quick and dirty’ paper, with experiments being carried out once. The authors show that GABARAP is expressed on/ in EV. That is a new finding that should be explored more: what type of EVs are these? Might they be exosomes/ microvesicles? Or do they express lysosomal proteins? Are they secretory lysosomes? Or autophagosomes? This brings me to the point of the lack of characterization of the secreted EV. In most figures, no or a very minimal EV control markers are shown. Furthermore, the blots that ‘show’ expression of endogenous GABARAP expression in EV preparations are really poor; based on this blot I don’t believe GABARAP is in or on these EVs. Although, APEX2-GABARAP seems to end up in isolated EV, but, keep in mind that most over expressed proteins end up in EVs, so to me that is not really convincing. 

Nevertheless, I like the concept of APEX2 proximity labeling method which is a nice way to determine a ‘co-secretome’ of a protein expressed in a EV(population). But to me it is unclear why the authors chose GABARAP to demonstrate this technique in EVs.

More detailed comments below:

  1. In figure 1a the authors want to show that GABARAP is expressed on EVs isolated from 3 different cell lines. I have some concerns with this figure. Mainly the lack of characterization of EVs and EV preparations. Controls and EV markers are missing (see MISEV guidelines). Basically the authors do not show at all that they actually isolated EVs here. How are the EVs isolated? I wonder why the authors used exoquick for these experiments, since they are able to do optiprep gradient centrifugation (1b), which is a more excepted method in the field.
  2. I am not convinced by the GABARAP expression in most preparations, which is extremely low, and is difficult to distinguish from background bands/ impurities(fig 1b). The authors should consider other techniques to determine GABARAP expression of EVs. (ELISA, EM i.e)
  3. In figure 1B, EVs are isolated through optipep density centrifugation. Also here, keep in mind the MISEV guidelines. Only ALIX as a EV marker is insufficient. Better characterization of EVs is required. The main purpose of this figure is to show GABARAP expression in EV preparations. Indeed, with some fantasy, there is a positive band on this blot, but the blot is of insufficient quality to determine that. The discrepancy between GABARAP expression with optipep gradient centrifugation and exoquick preprations is worrying.
  4. The absence of EV control markers (such as tetraspanins, or lysosomal/ autophagosomal proteins to distinguish between exosomes, secretory lysosomes, autophaosomes etc) don’t tell me these are really EVs. Keep in the MISEV guidelines in mind. Since GABARAP is associated with autophagosomes and lysosomes, these related markers should give more information about the character of these vesicles.
  5. In order to test the integrity of EVs, the authors show TEM micropgraphs. Based on these micrographs, one cannot determine if these are intact EVs or EVs at all.
  6. In line 268-270. Thus, the EV sample was characterised with an additional method next to TEM and immunoblotting of EV markers in accordance with the “Minimal information for studies of extracellular vesicles 2018 (MISEV 2018)” I can not find this additional method in the figure.
  7. Figure 1 C. the blot of ALIX in cell lysates is missing.
  8. In figure 1 D the authors show that GABARAPL1 is expressed on EVs isolated from plasma. However the expression seems to be extremely low. I am not convinced by this GABARAP blot.
  9. In figure 2 A, the authors show that endogenously expressed GABARAP and APEX2-GABARAP have a similar distribution. Please also add a control with endogenously GABARAP in cells not expressing APEX2-GABARAP (wild-type/ parental cells) to show GABARAP distribution. Without this control, this figure doesn’t tell the whole story.
  10. In line 378 the authors state that 50% of proteins are also found in the autophagosomal GABARAP-proxitome. This brings me to the point that EVs that contain APEX2-GABARAP are not well characterized. These EVs could also be secreted autophagosomes or lysosomes. How do the authors address this?
  11. The APEX2 labeling experiments are done without nutrient starvation to minimize stress related artefacts. (line 337). Apparently the authors are aware of stress-related artefacts that can occur during these conditions, so why are the experiments to determine endogenous GABARAP carried out during starving conditions in figure 1?
  12. A table of the proxitome would be more informative.
  13. The proxitome of GABARAP of secreted EVs overlaps with autophagosomal GABARAP. This brings me to one of the earlier points: maybe the authors are looking at secretory autophagy. It would be more informative if the secreted EV were characterized better.
  14. MS data always gives a lot of information which is sometimes hard to make it understandable and exiting to read. This is also the case in figure 5. To me, the figure does not add anything and is quite complicated to follow. All the graphs look equal and seem to be carried out once?

Reviewer 2 Report

Review Cells 750509

This article describes a new method to analyze the protein content of extracellular vesicles (EVs). In particular the analysis of proteins in close proximity of one protein of interest. As example, the authors studied the proxitome of the protein GABARAP in cells and EVs. This approach for specific analysis of proteins is new to the field of EVs and could give more insight into EV composition, especially in regard to the analysis of specific EV subpopulations that are defined by a certain protein marker. Also the potential association between autophagy and EV-secretory pathways is an interesting, new approach. Based on this novelty of the content of this article, I would recommend to accept this manuscript after minor revisions.

General comments:

  1. EV-isolation:
    1. Two isolation methods for EVs were given in the methods. What is the rational behind this? Which was used for which consecutive analysis? This is important since the method of isolation can affect the results of analysis. Please state this in the methods section.
    2. Furthermore the isolation methods used are rather unspecific and co-isolate a lot of contaminants like protein aggregates or other particles. Hence, it should be discussed in the manuscript that it can not be excluded that GABARAP and it’s proxitome is actually simply co-isolated with EVs, but not secreted via EVs. To show that the protein GABARAP is really found on or in EVs cryo electron microscopy with gold-immunolabelling could be a valid method to proof that vesicular structures with a lipid bilayer carry the protein of interest. If such methods can not be performed, this limitation should be discussed in the manuscript.
    3. In Addition to this, were there any further controls included, like supernatant of centrifugation, or isolation from unconditioned DMEM (because there might be FBS-EVs left). The performed density gradient purification could also help in separating protein aggregates from EVs. The proteomics/proxitomics analysis of several fractions of the density gradient (e.g. fraction 6 as no-EV control and 7-10 as EV samples) and comparison of the results could give more information on background protein signals.
  2. In general the number of replicates for each experiment should be given, i.e. biological replicates (how many different passages were tested) and if applicable technical replicates. The figure legend should then state that one representative image of n replicates is shown.
  3. Western Blot results, stainfree images: the term “stainfree” is a bit confusing to the reviewer. It seems that a general protein staining (ponceau, coomassie blue, etc.) on the membrane was performed? Please specify the procedure in the methods.
    Additionally, please explain why only small parts of the membranes (and why exactly these parts of the membranes) are displayed as “stainfree”. If the authors want to visualize the total protein amount, wouldn’t it make sense to show the whole gel/membrane, because only then you can get the full picture of protein loading per lane? Therefore the display of the whole membrane in the supplements would be sufficient in the eyes of the reviewer.
  4. Supplementary information: More detailed figure legends should be added for all supplementary figures, e.g. in a separate supplementary file. The supplementary figure legends should be as detailed as the legends of the figures included in the article and give all relevant information to understand the figure. For instance, they should include a scale bar description in the TEM figure legend, explanations for the different areas of the western blot membranes, what do the different colours mean,… The supplemental figures should be labelled consecutively, namely: S4A and S4B, as well as S5A and S5B. The reviewer understands the idea that these images are supporting the figures 2B and 2C or 3B and 3C, but since the figure number is not the same the letter should also be given in sequential order.

Minor comments to specific Lines:

  • Line 86: The examples of GABARAP functions/bindings should be given before making the statement that GABARAP is a “versatile binding hub”.
  • Line 129: after the name of the cell lines the word cells should be added: human embryonic kidney 293 CELLS, and hepatocellular carcinoma CELLS
  • Line 137: The half sentence “for IB and MS, poly D-Lysine coated bottom dishes” does not make sense at this position. Also it is not clear, whether the same cell numbers were used etc. Please specify.
  • Line 514: “an own mechanism” doesn’t sound like proper English. Please rearrange the sentence to avoid this wording

Round 2

Reviewer 1 Report

the authors adequately responded to all issues raised, I therefore recommend acceptance and publication of the manuscript